# Urinary Tract Infections: The Current Scenario and Future Prospects

**DOI:** 10.3390/pathogens12040623

**Published:** 2023-04-20

**Authors:** Giuseppe Mancuso, Angelina Midiri, Elisabetta Gerace, Maria Marra, Sebastiana Zummo, Carmelo Biondo

**Affiliations:** 1Department of Human Pathology, University of Messina, 98125 Messina, Italy; 2ASP (Azienda Sanitaria Provinciale), 90141 Palermo, Italy

**Keywords:** uropathogens, virulence factors, pathogenesis, antibiotic resistance

## Abstract

Urinary tract infections (UTIs) are among the most common bacterial infections worldwide, occurring in both community and healthcare settings. Although the clinical symptoms of UTIs are heterogeneous and range from uncomplicated (uUTIs) to complicated (cUTIs), most UTIs are usually treated empirically. Bacteria are the main causative agents of these infections, although more rarely, other microorganisms, such as fungi and some viruses, have been reported to be responsible for UTIs. Uropathogenic *Escherichia coli* (UPEC) is the most common causative agent for both uUTIs and cUTIs, followed by other pathogenic microorganisms, such as *Klebsiella pneumoniae*, *Proteus mirabilis*, *Enterococcus faecalis,* and *Staphylococcus* spp. In addition, the incidence of UTIs caused by multidrug resistance (MDR) is increasing, resulting in a significant increase in the spread of antibiotic resistance and the economic burden of these infections. Here, we discuss the various factors associated with UTIs, including the mechanisms of pathogenicity related to the bacteria that cause UTIs and the emergence of increasing resistance in UTI pathogens.

## 1. Introduction

The urinary system consists of the kidneys, ureters, bladder, and urethra, and its main function is to filter blood by removing waste products and excess water. The urinary system plays a key role in removing the waste products of metabolism from the bloodstream. Other important functions performed by the system are the normalization of the concentration of ions and solutes in the blood and the regulation of blood volume and blood pressure [1]. In healthy people, urine is sterile or contains very few microorganisms that can cause an infection [2]. Urinary tract infections (UTIs) are infections that can occur in the urethra (urethritis), bladder (cystitis), or kidneys (pyelonephritis) and are one of the world’s most common infectious diseases, affecting 150 million people each year, with significant morbidity and high medical costs (e.g., it has been estimated that the economic burden of recurrent UTIs in the United States is more than $5 billion each year) [3,4]. Although symptomatology varies depending on the location of these infections, UTIs have a negative impact on patients’ relationships, both intimate and social, resulting in a decreased quality of life [5,6]. UTIs are classified as either uncomplicated (uUTIs) or complicated (cUTIs) [7]. uUTIs typically affect healthy patients in the absence of structural or neurological abnormalities of the urinary tract [4]. cUTIs are defined as complicated when they are associated with urinary tract abnormalities that increase susceptibility to infection, such as catheterization or functional or anatomical abnormalities (e.g., obstructive uropathy, urinary retention, neurogenic bladder, renal failure, pregnancy, and the presence of calculi) [4,8].

In both community and hospital settings, the Enterobacteriaceae family is predominant in UTIs, and the main isolated pathogen is uropathogenic *Escherichia coli* (UPEC) [9,10]. The latter is also the most common causative agent of cUTI [10]. Antibiotic-resistant Gram-negative bacteria are more prevalent in hospitals than in community samples (e.g., carbapenemase-resistant Enterobacteriaceae) [11]. UTIs are mainly caused by bacteria, while the involvement of other microorganisms, such as fungi and viruses, is quite rare. *Candida albicans* is the most common type of fungus that causes UTIs. Common causes of viral UTI are cytomegalovirus, type 1 human Polyomavirus, and herpes simplex virus [12,13].

This review pursues a twofold goal: the first is to provide an overview of the mechanisms underlying the pathogenesis of UTIs; the second is to provide an overview of recent advances in new strategies, as an alternative to antibiotics, to control the spread of multidrug-resistant UTI isolates.

## 2. Pathogenesis of UTI

Urinary tract infections (UTIs) begin when gut-resident uropathogens colonize the urethra and subsequently the bladder through the action of specific adhesins. If the host’s inflammatory response fails to eliminate all bacteria, they begin to multiply, producing toxins and enzymes that promote their survival. Subsequent colonization of the kidneys can evolve into bacteremia if the pathogen crosses the kidney epithelial barrier (Figure 1). In complicated UTIs, infection by uropathogens is followed by bladder compromise, which occurs with catheterization. A very common situation is the accumulation of fibrinogen on the catheter as a result of the strong immune response induced by catheterization. Uropathogens, through the expression of fibrinogen-binding proteins, bind to the catheter. Bacteria also multiply as a result of biofilm protection, and if the infection is left untreated, it can progress to pyelonephritis and bacteremia (Figure 1). UTIs are the most common bacterial infection in humans worldwide and the most common hospital-acquired infection [14,15]. The spread of UTIs is closely linked to the effectiveness of a number of strategies that uropathogens have developed to adhere to and invade host tissues [16,17]. Often, the infection does not seem particularly severe, especially in the early stages, but it can worsen significantly in the presence of complicating factors [18,19]. Complicating factors that are involved in the progression of UTI are biofilms, urinary stasis due to obstruction, and catheters. UTIs comprise a heterogeneous group of clinical disorders that vary in terms of the etiology and severity of conditions. The risk of UTI is influenced by a wide range of intrinsic and acquired factors, such as urinary retention, vesicoureteral reflux, frequent sexual intercourse, prostate gland enlargement, vulvovaginal atrophy, and family history. The use of spermicides may also increase the risk of UTI in women [19,20]. A urine culture with ≥105 colony-forming units/mL without any specific UTI symptoms is defined asymptomatic bacteriuria, as it usually resolves spontaneously and does not require treatment [21]. Asymptomatic UTIs should be treated only in selected cases, such as pregnant women, neutropenic patients, and those undergoing genitourinary surgery, as antibiotic treatment may contribute to the development of bacterial resistance [22,23]. In contrast, symptomatic UTIs are commonly treated with antibiotics that can alter the intestinal and vaginal microbiota, increasing the risk factors for the spread of multidrug-resistant microorganisms [4,23,24]. 

## 3. Classification of UTI

In general, UTIs are named according to the site of infection: urethritis is inflammation of the urethra, ureteritis refers to inflammation of the ureter, and cystitis and pyelonephritis involve the bladder and kidney, respectively [25]. UTIs are further classified according to the presence of predisposing conditions for infection (uncomplicated or complicated) or the nature of the event (primary or recurrent) [25,26,27]. In most cases, uUTIs are caused by uropathogens that reside in the intestine and, after accidental contamination of the urethra, migrate, colonizing the bladder [28]. While sharing the same dynamics described for uncomplicated infections, cUTIs occur in the presence of predisposing factors, such as functional or structural abnormalities of the urinary tract [29]. Other typical features of complicated UTIs include a significantly higher rate of treatment failure and systemic or invasive tissue involvement [22,28]. Three or more uncomplicated UTIs within 12 months or two or more infections within six months define recurrent UTIs; usually, recurrences in this type of infection are due to the same microorganism responsible for the previous infections [30,31].

## 4. Immune Response to Uropathogens

Although the urinary tract is often exposed to microorganisms from the gastrointestinal tract, infection by these microorganisms is a rather rare occurrence due to the innate immune defenses of the urinary tract [32]. Previous studies have shown that the immune response is carefully regulated so as not to compromise the structural integrity of the epithelial barrier. Macrophages and mast cells play a key role in immune regulation of the urinary tract, coordinating the recruitment and initiation of neutrophil responses that lead to the removal of bacteria in the bladder. In addition, these cells are critical in preventing an excessive neutrophil response from causing damage to bladder tissue and predisposing this organ to persistent infection [33].

## 5. Virulence Factors of the Main Uropathogens

The ability of different uropathogens to successfully adhere to and colonize the epithelium of the lower urinary tract is related to their ability to express specific virulence factors [34]. Uropathogenic *E. coli* (UPEC) is the most common causative agent of both uUTIs and cUTIs [34]. Most UTIs are caused by Gram-negative and Gram-positive bacteria residing in the colon, such as *Escherichia coli*, *Enterococcus faecalis*, *Proteus mirabilis*, and *Klebsiella pneumoniae* [4,34]. Other causative agents include *Staphylococcus saprophyticus, Group B Streptococcus (GBS),* and *Pseudomonas aeruginosa* [4]. Figure 2 shows the epidemiology of different uropathogens in uUTIs and cUTIs. On the cell surface of uropathogens are several adhesion proteins that play a crucial role in the initial interactions between the host and pathogen [34]. In addition, adhesins have recently been found to promote both the attachment of bacteria and invasion of host tissues in the urinary tract. Among the best-known adhesion factors are the pili of uropathogenic bacteria belonging to both Gram-positive and Gram-negative bacteria. Two distinct pathways are required for pili assembly in Gram-negative and Gram-positive bacteria, known as the chaperone/usher pathway and the sortase-assembled pili pathway, respectively [34]. These uropathogens use different types of adhesins that promote binding and biofilm formation on biotic and abiotic surfaces. In this context, it is important to note that most UTIs are biofilm-associated infections in which uropathogens colonize both the mucosa of the urinary tract and indwelling devices such as urinary catheters [35]. Biofilm formation by these pathogenic bacteria requires specific virulence factors that play a key role in inducing adhesion to host epithelial cells or catheter materials [35,36]. Bacterial biofilms play an important role in UTIs, being responsible for the persistence of infections that result in recurrence and relapse. Since eradication of biofilms often cannot be achieved by antibiotic treatment, new approaches for eradication of aggressive biofilms are being tested, such as phagotherapy, enzymatic degradation, antimicrobic peptides, and nanoparticles [37]. Table 1 shows the main types of adhesins that are crucial for biofilm formation. These structures promote the attachment of uropathogens to biotic/abiotic surfaces. Uropathogens are also producers of toxins, proteases (e.g., elastase and phospho-lipase) and iron-harvesting siderophores, all of which are involved in the onset and spread of UTIs [38].

### 5.1. UPEC

UPEC is the leading cause of UTIs and is responsible for at least 80% of community-contracted UTIs and 65% of hospital-contracted UTIs [48,49]. Although UPEC strains are present in the human intestinal tract, they differ from commensal strains of *E. coli* in their ability to express a multitude of virulence factors that allow their transit from the intestinal tract to the urinary tract following fecal contamination of the periurethral area [40,50]. Although virulence factors, such as toxins, surface polysaccharides, flagella, and iron acquisition systems, are important in overcoming host defenses and establishing urinary infection, adhesion of UPECs to host epithelial cells remains the most important determinant of pathogenicity [18,34].

To successfully colonize the urinary tract, UPECs must be able to adhere to host cells, colonize the urethra, adhere to the surface of the bladder epithelium, and, in some cases, form biofilms with the creation of bladder intracellular communities (IBCs) [34,51]. UPEC infection elicits innate immune responses characterized by the production of different proinflammatory cytokines and chemokines [34]. The host’s inflammatory response leads to the rapid recruitment of neutrophils into the bladder lumen and exfoliation of infected bladder epithelial cells [52]. UPEC escapes the host innate immune response by taking refuge in the cell cytoplasm, where it can multiply rapidly by forming IBCs [18]. This condition allows both bacterial invasion of other host cells and the re-entry of UPECs into the IBC cycle. At the same time, UPECs remain viable for a long time within quiescent intracellular compartments [34]. These structures, located in the underlying transitional cells, contain some viable non-replicating bacteria (usually less than 10). These can reactivate, causing recurrent urinary tract infections [18,32]. UPEC strains express a broad spectrum of virulence factors, but their ability to cause UTIs is fundamentally related to their ability to produce a number of adhesins that can facilitate adhesion under different environmental conditions. These factors are also crucial for the survival of this microorganism [53]. The main virulence factors involved in host cell adhesion are Type 1 and Type 2 fimbriae, P fimbriae, Dr adhesion, S fimbriae, and F1C fimbriae [40]. Type 1 fimbriae, via the adhesin subunit FimH, located at the terminal end of the fimbriae, bind to the urothelial uroplakin, facilitating biofilm formation during UTI [53]. Type 2 fimbriae are another important virulence factor of UPECs, which, through recognition and binding to glycosphingolipids, play an important role in the pathogenesis of ascending UTI and pyelonephritis [18,40]. Dr adhesion and S fimbriae mediate the attachment of this pathogen to uroepithelial cells of the kidney and to sialic acid molecules located in urothelial cells of the bladder [54]. F1C fimbriae are present only in the absence of F pili and P pili and bind to specific receptors present in endothelial cell lines of the lower urinary tract and in the kidneys [18,36]. In addition to the above surface virulence factors, UPECs also produce secreted virulence factors, the most important of which are HlyA, a lipoprotein called α-hemolysin, associated with the most severe UTIs, and CNF1 (necrotizing cytotoxic factor 1) involved in pyelonephritis and renal invasion [55]. HlyA is a pore-forming toxin that exerts a dual, concentration-dependent effect on renal epithelial cells. At high concentrations, HlyA is capable of damaging host cells, thereby facilitating iron release and nutrient acquisition. At low concentrations, HlyA can induce apoptosis of target host cells, thereby promoting the spread of UPEC to other host cells. Necrotizing cytotoxic factor 1 (CNF1) functions by binding to the basal cell adhesion molecule receptor (BCAM) and inducing constitutive activation of RHO GTPases. Activation of the latter promotes increased levels of bacterial internalization and the spread of UPEC to other host cells [4].

### 5.2. K. pneumoniae

*K. pneumoniae* has emerged as a major cause of healthcare-associated opportunistic infections, such as bacteremia, pneumonia, and UTIs [56]. This pathogen, similar to UPEC, uses two types of fimbrial adhesin, type 1 fimbriae and type 3 fimbriae, for biofilm formation and bladder colonization [44]. Notably, these two types of fimbriae adhesins have different binding specificities. Type 1 fimbriae bind to mannose receptors in the urinary tract and promote bladder cell invasion [34]. Type 3 fimbriae, on the other hand, do not bind to mannose receptors but play an important role during biofilm aggregation on medical devices such as catheters [4,34].

### 5.3. P. mirabilis

Due to the production of different fimbriae, such as mannose-resistant Proteus fimbriae (MRP fimbriae), *P. mirabilis* is the most commonly identified Gram-negative bacterium in cUTIs, especially in catheterized patients or patients with urinary tract abnormalities [57]. Other pili produced by *P. mirabilis* are *P. mirabilis*-like fimbriae (PMF) and nonagglutinating fimbriae (NAF), which are involved in bladder and kidney colonization and uroepithelial cell adhesion, respectively [58]. In addition, adhesion and invasion of *P. mirabilis* into the bladder and kidney are mediated by two autotransporters, TaaP (Proteus autotransporter trimeric) and AipA (Proteus autotransporter-mediated adhesion and invasion), which are able to bind collagen I and laminin, respectively [59]. A key role in catheter-associated urinary tract infections (CAUTI) caused by *Proteus mirabilis* lies in the ability of this pathogen to produce urease, a Ni2+-dependent metalloenzyme that hydrolyzes urea into carbon dioxide and ammonia [60]. The resulting increase in urine pH induces the formation of calcium crystals and precipitates of magnesium and ammonium phosphate, which makes possible the formation of a crystalline biofilm on the catheter that protects this pathogen from the host immune system and antibiotics. *P. mirabilis* urease also plays an important role in the formation of stones that prevent proper urine drainage, causing reflux and promoting the progression of infection to pyelonephritis and septicemia [61]. In addition, this rod-shaped bacterium produces two toxins, hemolysin (HpmA) and Proteus toxic agglutinin (Pta), both of which are essential for disruption of host tissues and spread of the bacterium to the kidneys, resulting in acute pyelonephritis [61]. These toxins also play an important role in bacterial infection, related to the release of nutrients following lysis of the host cell. The ability to utilize these substances is critical for bacterial replication (e.g., iron recovery through siderophores) [4,59].

### 5.4. Enterococci

Enterococci cause several nosocomial infections, particularly surgical site/soft tissue infections, bloodstream infections, and UTIs [62,63]. These uropathogens do not contain pili and adhere to the host cell via their surface proteins, such as Esp (Enterococcal Surface Proteins) and Ebp (Endocarditis and biofilm-associated pilus) [64]. Catheter-associated urinary tract infections (CAUTI) caused by Enterococci are due to the release of fibrinogen in the bladder after urinary catheterization. EbpA, which contains an N-terminal fibrinogen-binding domain, binds to fibrinogen deposited on the implanted catheter and promotes the formation of the biofilm responsible for *E. faecalis* CAUTI [65]. In addition, biofilm formation on the catheter helps bacteria evade the host immune system by masking antigenic determinants [4,28].

### 5.5. S. saprophyticus

*S. saprophyticus* is a coagulase-negative microorganism responsible for uUTIs, such as cystitis, in sexually active women [66]. This Gram-positive bacterium is the second most common cause of community-acquired UTI, after *E. coli*. Bacterial adhesion to the bladder and ureter epithelium occurs through different types of adhesins, such as Aas, Uaf, and SdrI. These include cell wall-associated proteins with hemagglutinic and adhesive properties, as well as surface-associated glycoproteins that facilitate bacterial adhesion to the host cell surface and promote bladder colonization [34]. Moreover, as in *Proteus mirabilis* and *Klebsiella pneumoniae* but not in *E. coli*, *S. saprophyticus* urease is responsible for persistent bacterial colonization in the bladder and kidney [4]. This enzyme, which catalyzes the hydrolysis of urea into carbon dioxide and ammonia, causes an increase in urine pH and the production of carbonate precipitates (stone formation) in the urine [66].

### 5.6. P. aeruginosa

Within the hospital setting, *P. aeruginosa* is the third most common cause of urinary tract infection (7−10%) after *E. coli* and *E. faecalis* [63]. Patients with underlying conditions, such as urinary tract abnormalities or indwelling urinary catheters, are more susceptible to UTI caused by *P. aeruginosa* [67]. The inherent multiple antibiotic resistance of this microorganism, combined with its ability to develop new resistance to multiple classes of antibiotics and to form biofilms, explains the high mortality and morbidity of UTIs caused by *P. aeruginosa* [67]. This microorganism has the ability to form biofilms on catheters through the production of various components such as extracellular polysaccharides, elastase, exoenzyme S, and hemolytic phospholipase C. Elastase is a major virulence factor of *P. aeruginosa*. This enzyme, through its protease activity, induces tissue destruction by releasing nutrients essential for bacterial growth. ExoS is present in invasive strains of *P. aeruginosa* and acts on actin cytoskeleton rearrangement, affecting cell adhesion, morphology, and apoptosis in target host cells. Phospholipase C acts by hydrolyzing phosphatidylcholine from the host cell membrane, causing cellular damage and organ failure. All these factors are regulated by the quorum sensing system and are involved in the spread of UTI to the kidneys, resulting in pyelonephritis [68].

## 6. Diagnosis and Treatment of UTI

A bacterial count greater than or equal to 100,000 CFU/mL is considered diagnostic of UTI, although this value results in a large number of false negatives that fail to detect many relevant infections [69]. Previous studies have shown that patients with symptomatic UTIs can have bacterial counts as low as 103 cfu/mL [70]. Bacteriuria, or the presence of bacteria in urine without symptoms, is not an infection and should be treated only in exceptional cases, such as in pregnant women or before any urologic procedure [71] (Figure 3). In addition, although international guidelines recommend that cultures with more than one microbial species should be considered contaminated (i.e., urine specimens were not collected midstream), it should be noted that many UTIs are polymicrobial, especially those affecting the elderly population, catheter-associated urinary tract infections, and cUTIs [72]. Previous studies have shown that for patients with recurrent UTI or UTI symptoms who have tested negative on standard urinoculture, an additional tool known as extended quantitative urinoculture can be used. This test allows for better identification of fastidious or slow-growing bacteria than standard urinoculture, as it involves higher plate volumes and incubation times than the standard method [73]. In addition, although still uncommon in clinical laboratories, protocols, and technologies, such as flow cytometers, mass spectrometry, and multiplex PCR panels, are now available that can identify pathogens very rapidly by directly analyzing clinical urine samples [74,75]. Moreover, new technologies are emerging, such as biosensors, microfluidics, and real-time microscopy platforms that, directly from clinical urine samples, can identify the pathogen and its susceptibility to antibiotics [75]. Treatment of asymptomatic bacteriuria is not recommended because it increases the risk of symptomatic UTI and contributes significantly to future resistant infections [76]. International guidelines recommend three options for first-line treatment of acute uncomplicated cystitis: fosfomycin, nitrofurantoin, and pivmecillinam. Trimethoprim/sulfamethoxazole could be considered a first-choice drug but only if local resistance to *Escherichia coli* does not exceed 20 percent. Aminopenicillins and fluoroquinolones are no longer recommended as first-line therapies for urinary tract infections because of high resistance rates and potentially long-lasting side effects, respectively. Second-line options include oral cephalosporins, such as cephalexin or cefixime, fluoroquinolones, and β-lactams, such as amoxicillin-clavulanate. Recurrent UTIs are common. The prevention of UTIs consists of risk factor avoidance, non-antimicrobial measures, and antimicrobial prophylaxis. The main risk factors associated with UTI recurrence are related to low estrogen levels (i.e., reduced numbers of beneficial lactobacilli), diabetes, urinary incontinence, vaginal wall prolapse, and incomplete bladder emptying. Pyelonephritis is an infection of the upper urinary tract, and fever, chills, nausea, costovertebral angle tenderness, and vomiting are the most common signs and symptoms. It is important to distinguish between uncomplicated and complicated pyelonephritis, as the management and disposition of patients are completely different [38]. Oral fluoroquinolones are recommended as first-line agents for uncomplicated pyelonephritis. Other acceptable agents, if fluoroquinolones cannot be used, are trimethoprim-sulfamethoxazole or beta-lactams. Complicated obstructive pyelonephritis requires intravenous antibiotic treatment, as it can rapidly lead to urosepsis. Ceftolozane/tazobactam or ceftazidime-avibactam combinations have proven effective for the treatment of UTI from resistant Enterobacterales and *Pseudomonas aeruginosa*. A new class of drugs includes imipenem/cilastatin, cefiderocol, meropenem-vaborbactam, and plazomycin [66]. These new agents could provide a viable alternative in the treatment of complicated infections resistant to carbapenems. The clinical management of cUTI depends on the severity of illness at presentation. Patients should be treated initially with an intravenous antimicrobial regimen, such as amoxicillin plus an aminoglycoside, a second-generation cephalosporin plus an aminoglycoside, or a third-generation cephalosporin with or without an aminoglycoside. Alternative treatments of cUTIs caused by multidrug-resistant pathogens include the following combinations: Ceftolozane/tazobactam, Imipenem/cilastatin, and ceftazidime/avibactam.[77]. However, because bacteria have developed different antibiotic resistance mechanisms, it is essential to perform an antibiotic susceptibility test to determine which antibiotic will be most effective in treating the infection. For example, a combination of monobactams and two β-lactamase inhibitors is effective against many carbapenemase-resistant *Enterobacteriaceae*, but not against *K. pneumoniae* strains harboring ESBL, AmpC, and carbapenemase genes simultaneously [78,79]. Due to the indiscriminate and widespread use of antibiotics, both the increase in antibiotic resistance and the recurrence rates of infections caused by these uropathogens have reached alarming levels [4]. According to the latest Center for Disease and Control (CDC) report, the impact of antibiotic-resistant infections is estimated at 2.8 million antibiotic-resistant people and 35,000 deaths each year in the United States [27,79,80]. Most of the deaths were caused by six AMR pathogens: *Escherichia coli*, followed by *Staphylococcus aureus, Klebsiella pneumoniae, Streptococcus pneumoniae, Acinetobacter baumannii, and Pseudomonas aeruginosa* [27]. Although phage therapy has many advantages over antibiotic therapy, such as host specificity, prevention of biofilm formation, and few side effects, both the narrow host range and the emergence of phage-resistant strains limit its use for the treatment of drug-resistant UTIs [75,80].

## 7. Antimicrobial Resistance in UTIs

UTIs are mainly caused by Gram-negative bacteria that are becoming an increasing threat to public health because of their ability to acquire genes, located on transferable plasmids, that code for extended-spectrum β-lactamases (ESBLs) [75]. These enzymes are capable of hydrolyzing third-generation cephalosporins and monobactams but not carbapenems [81]. In addition, ESBLs pose a public health problem because they are encoded on plasmids that usually carry other resistance genes against different classes of antibiotics (e.g., aminoglycosides, sulfonamides, and quinolones) [82,83]. As a result, bacteria that acquire these plasmids become multidrug resistant. Although all ESBLs function through cleavage of the amide bond of the β-lactam ring, the genes encoding these enzymes are diverse and grouped into different families [84]. Whereas until 2000, TEM- and SHV-type ESBLs, characterized by their ability to hydrolyze extended-spectrum β-lactam antibiotics and inhibition by β-lactamase inhibitors, such as clavulanate, tazobactam, and avibactam, were the predominant ESBL families, today the most commonly encountered ESBL types are phylogenetically distinct from the first β-lactamases that appeared in the early 1980s [85]. CTX-M type enzymes are the most commonly encountered ESBL types, being present in several members of the order Enterobacterales in *P. aeruginosa* and *Acinetobacter* spp. [79]. Isolated strains carrying CTX-M confer high-level resistance to cefotaxime and have reduced susceptibility to ceftazidime [86]. Other types of ESBLs are OXAs, AmpCs, and Carbapenemases [4]. Oxas and AmpC are β-lactamase enzymes encoded by chromosomal and plasmid genes that resist inhibition by β-lactamase inhibitors [87]. *K. pneumoniae* carbapenemase (KPC) and New Delhi metallo-β-lactamase (NDM-1) are enzymes that make Enterobacteriaceae resistant to a wide range of beta-lactam antibiotics, particularly carbapenemases (CRE) [88]. Other important mechanisms of resistance are limitation of absorption of a drug, modification of a drug target, and active efflux of a drug. Some bacterial proteins are targets of antimicrobials. Alteration of these bacterial proteins so that the drug binds poorly or does not bind at all is a common mechanism of resistance. The most common mechanism of bacterial resistance is the efflux of drugs from cells through membrane transporters. These transporters are proteins that belong to a superfamily of genes called the ATP-binding cassette (ABC). Overexpression of ABC transporters is a major determinant of multidrug resistance, as it increases the efflux of different drugs from cells, thereby decreasing the intracellular concentration of drugs [89]. The number of community- and hospital-acquired urinary tract infections is steadily increasing due to the growing resistance of uropathogens to antibiotics. Isolation rates of fluoroquinolone-resistant Enterobacteriaceae have increased to such an extent that they are no longer recommended as the empiric therapy of first choice [88,90]. Since plasmids that code for ESBLs often also code for resistance to trimethoprim sulfamethoxazole, the latter is recommended as a first-line antibiotic for UTI only when local resistance rates do not exceed 20 percent [91]. More recently, fosfomycin, discovered more than 40 years ago, has been shown to be active against a wide range of ESBL-positive uropathogens and could be a viable therapeutic option against UTIs compared with ceftriaxone or meropenem [92]. In addition to the dominant resistance mechanisms described above for ESBLs belonging to the CTX-M, TEM, and SHV families, there are a multitude of different resistance mechanisms among uropathogens that are more or less widespread depending on the local epidemiological context [89]. Although UTIs are mainly caused by Gram-negative bacteria, Gram-positive bacteria, such as *Staphylococcus aureus* (MRSA) and vancomycin-resistant *Enterococcus faecalis* (VRE), have emerged as important causative agents of UTIs, particularly among pregnant women, the elderly with high associated comorbidities, and catheterized patients [72]. In addition, enterococci exhibit intrinsic resistance to the most common antibiotics, such as cephalosporins, aminoglycosides, clindamycin, and trimethoprim-sulfamethoxazole [93]. Several strategies have been tried to prevent or treat infections with these resistant pathogens, including combinations of antibiotics, antimicrobial peptides, and bacteriocins. In the treatment of antibiotic-resistant bacteria, the interpretation of susceptibility patterns also depends on the clinical situation and the availability of therapeutic options. For example, the concentration of gentamicin obtained in urine may be high enough to treat a lower urinary tract infection caused by a microorganism identified as resistant to gentamicin [79]. Recently, there has been considerable interest in combinations of antibiotics, such as tigecycline and Fosfomycin [94]. The most promising approaches for the treatment of cUTI involve antibiotic-inhibitor combinations, such as ceftazidime/avibactam (combination of a third-generation cephalosporin with a next-generation β-lactamase inhibitor, such as avibactam) [95,96]. Treatment should be guided by local susceptibility profiles and antibiogram results.

## 8. Advances in the Management of Antibiotic Resistant UTI

UTIs are responsible for a large number of antibiotic prescriptions, which are known to be a major cause of the spread of antimicrobial resistance [26,79]. Therefore, finding new drugs to combat antimicrobial resistance and expanding the field of research to find new treatment options have become top priorities. A recent study published by Söderström et al. described for the first time how UPEC bacteria spread and multiply [97]. Using a human bladder cell infection model, the authors found that during the infection cycle of UTIs, UPECs form spaghetti-like filaments that measure several hundred times their original length before reverting to the rod shape [97]. Although further studies are needed to clarify why the bacteria perform this transformation, this study has paved the way for the discovery of new therapeutic options for the treatment of UTIs caused by UPECs. This is very important considering that almost all UTIs (80%) are caused by uropathogenic *E. coli* [40]. Although the toxins and proteases of several uropathogens have been tested as potential vaccine targets for UTI prevention, further studies are needed to determine the efficacy of these vaccines [4]. Because uropathogens require an iron source during colonization, several siderophore systems have been studied as targets for vaccine development. These studies have shown that these siderophore vaccines are able to reduce bacterial colonization of the bladder in mice during infection and thus are valuable antigens to evaluate in future studies [98]. Although several vaccines have been investigated for the prevention of urinary tract infections, to date they have had little success, and no effective vaccines against urinary tract infections are currently available. A new strategy was recently reported in a study published by Wu et al. [99]. The authors administered the vaccine, combined with an adjuvant, directly into the bladder in a way that would increase the recruitment of bacterial elimination cells and prevent future infections [99]. New antibiotics are being developed for the treatment of UTIs, the most promising of which are gepotidacin and two oral carbapenems: tebipenem and sulopenem. Gepotidacin is a compound belonging to the pyranopyridine class that selectively inhibits bacterial DNA replication, while tebipenem and sulopenem are in various stages of clinical development for the treatment of complicated and uncomplicated UTIs [100,101].

## 9. Discussion and Conclusions

Men and women of any age can have urinary tract infections, but the incidence of urinary tract infections is higher in women than in men because of the female anatomy [102]. Most patients attending outpatient clinics complaining of dysuria have a UTI, although it is possible that patients presenting with symptoms of a UTI are instead suffering from overactive bladder or interstitial cystitis [103]. Diagnosis is not always straightforward. For many decades, midstream urinoculture has been considered the gold standard for UTI diagnosis. However, in about one-third of cases, a positive culture is not obtained, and it has become increasingly clear that bacteria may be present in the healthy bladder [104]. The impact of UTIs on individuals is significant, as infections negatively affect individuals’ mental health and sense of well-being [6]. In addition, patients with recurrent UTI due to treatment failure caused by antimicrobial-resistant strains have a reduced quality of life [6]. In this regard, several studies have documented resistance to cephalosporins commonly used to treat UTIs [77]. The implementation of good antimicrobial stewardship is critical to preventing the development of resistance and improving patient outcomes. The goal of antimicrobial stewardship is threefold and includes the implementation of specific strategies. The first goal is to prevent the treatment of asymptomatic bacteriuria; the second goal is to prevent the use of broad-spectrum fluoroquinolones; and the third goal is to minimize the development of resistance by adhering to recommended drug cycles and dosages [105]. According to recent studies, the elderly have an increased risk of contracting uncomplicated urinary infections that are resistant to multiple antibiotics [106]. The use of empirical antibiotics should be limited to cases where symptoms are unbearable and/or a more serious infection is feared [83]. The antibiotic to be prescribed must take into account local patterns of resistance to uropathogens and, of course, the patient’s possible allergies to antibiotics. Recent studies suggest that the urinary microbiota, in addition to its known beneficial role in maintaining bladder homeostasis, also plays a protective role against infection by forming a physical barrier [107]. In this regard, the current management of recurrent UTI involves repeated courses of antibiotics, which can change the balance of *Lactobacillus* spp. in the gut and bladder [107]. In healthy women, *Lactobacillus* deficiency has been associated with colonization of uropathogens, such as *E. coli*, *Klebsiella pneumoniae*, and *Pseudomonas aeruginosa*, which are responsible for recurrent UTIs [108]. The beneficial effect of the microbiome on UTI has been further demonstrated by the fact that women with bacterial vaginosis due to an overgrowth of *Gardnerella vaginalis* have a much higher risk of rUTI than healthy women with a microbiome represented by different species of *Lactobacillus* [107]. From the above, it is clear that the composition of the vaginal microbiome plays an important role in its susceptibility to recurrent UTI [109,110]. While no evidence supports the use of antibiotic prophylaxis for recurrent UTI, in contrast, there is increasing evidence in favor of nonantibiotic prophylaxis regimens for recurrent UTI [111]. In addition, because overuse of antibiotics is a major factor in the development of MDR bacteria and because about 25 percent of all antibiotic prescriptions are for UTIs, antibiotic prophylaxis should be used once all nonantibiotic treatment options have been exhausted [27]. The most recommended nonantibiotic prevention and treatment options for recurrent UTIs include cranberries, intravaginal probiotics (*L*. *rhamnosus*, *L*. *reuteri*), D-mannose, hippurate methenamine, estrogen-releasing vaginal ring in postmenopausal women, and immunostimulants [112]. Vaccine therapy has emerged as a promising alternative to antibiotics for the treatment and prevention of UTI [113]. A sublingual vaccine consisting of inactivated whole bacteria has been shown to be effective in reducing UTI recurrences nine months after starting treatment with the vaccine. Although the exact protective mechanism by which this vaccine reduces UTI recurrences is still not entirely clear, several authors suggest that the reduction in UTI recurrences is due to an enhancement of local innate immune mechanisms [99]. In addition, recent studies have shown the potential of bacteriophage therapy for the treatment of urinary tract infections caused by MDR bacteria, such as *E. coli* and *K. pneumoniae* [114]. However, although the preliminary data obtained from this therapy are very promising, there is still much preclinical and clinical work to be done before bacteriophages can be an alternative to antibiotics in the future.

## Figures and Tables

**Figure 1 pathogens-12-00623-f001:**
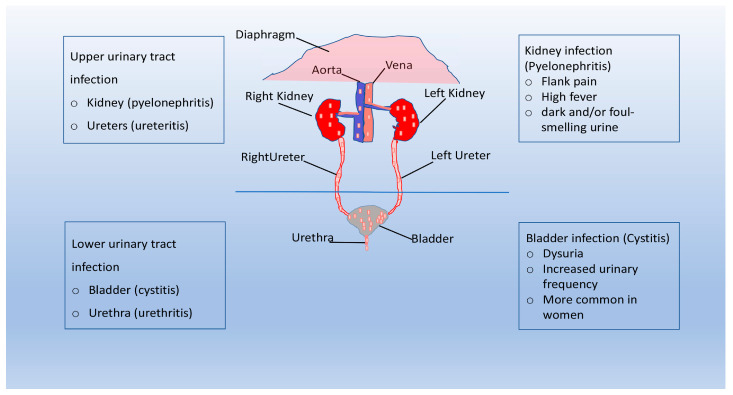
Pathogenesis of UTI. Urinary tract infections (UTIs) start when uropathogens colonize the urethra and subsequently the bladder through the action of specific adhesins. If the bacteria are able to evade the immune system, they begin to multiply and biofilms form. Bacteria can reach the kidney from the lower urinary tract, and UTI can evolve into bacteremia. In complicated UTI, uropathogens are usually able to bind to the catheter and multiply due to the protection of the biofilm. If left untreated, the infection can progress to pyelonephritis and bacteraemia.

**Figure 2 pathogens-12-00623-f002:**
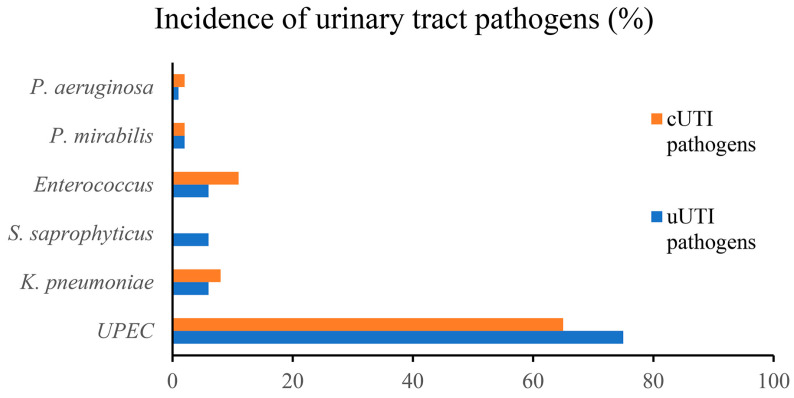
Epidemiology of uropathogens in uUTIs and cUTIs.

**Figure 3 pathogens-12-00623-f003:**
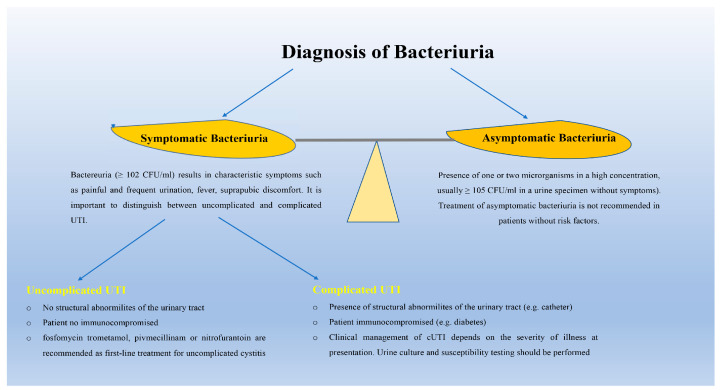
Diagnosis and management of bacteriuria. Asymptomatic bacteriuria does not result in urinary tract infections and does not require antibiotic treatment, which should be evaluated only in pregnant women or in a subject before undergoing urologic surgery. In healthy patients, uropathogens originate from the rectal flora and enter the bladder via the ascending route. Urinary tract infections can also occur via hematogenous or lymphatic routes (uncommon routes). UTIs can be classified as complicated or uncomplicated based on the presence of risk factors, such as age, catheterization, diabetes mellitus, comorbidities in pediatric patients, and spinal cord injury.

**Table 1 pathogens-12-00623-t001:** Main types of adhesins expressed by Gram-negative and Gram-positive uropathogens.

Uropathogens	Adhesin	Biotic/Abiotic Surface	References
*E. coli* (UPEC)	Type 1 fimbriae Type 2, P fimbriae Dr adhesion S fimbriae F1C	binds to kidney cells and promotes the formation of a biofilm binds to Globosides, a sub-class of the lipid class glycosphingolipid. binds bladder and kidney epithelial cells binds to receptors containing *sialic acid* binds to glycolipid receptors present in the endothelial cells of bladder and kidney and promotes biofilm formation	[39] [40] [41] [42] [43]
* K. pneumoniae *	Type 1 fimbriae Type 3 fimbriae	Binds to the mannose-binding receptors and promote *biofilm* formation on abiotic surfaces promote biofilm formation on abiotic surfaces	[44] [44]
* P. aeruginosa *	T4Pa	Binds to glycosphingolipid receptors present in host epithelial cells and promotes biofilm formation.	[34]
* P. mirabilis *	MRP fimbriae	binds mannosylated glycoproteins of bladder cells	[45]
	NAF fimbriae Mrp/H	binds with glycolipids promote the formation of biofilms in the urinary tract	[45,46]
* S. saprophyticus *	Aas adhesin SdrI adhesin Uaf adhesin	binds to human ureters binds to collagens binds to bladder epithelial cells	[34]
* E. faecalis *	Enterococcal Surface Protein	promote primary attachment and biofilm formation on biotic and abiotic surface	[47]

UPEC: UroPathogenic *Escherichia coli*; F1C, type 1-like immunological group C pili; T4Pa, type IV pili; MRP, mannose-resistant Proteus fimbriae; NAF, Non-agglutinating fimbriae; Aas, autolysin/adhesin of *Staphylococcus saprophyticus*; SdrI, *serine*-aspartate repeat proteins; Uaf, Uro-adherence factor.

## Data Availability

Not applicable.

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
