# Peer review of "Urinary Tract Infections: The Current Scenario and Future Prospects"

_pathogens, 2023, doi:10.3390/pathogens12040623_

Round 1

Reviewer 1 Report

The manuscript Urinary tract infections: the current scenario and future prospects by Mancuso and colleagues gives an important panorama of UTIs, describing general aspects of disease, virulence factors of the main urobacteria, diagnosis and therapeutic approaches.

Although articles previously published (Klein RD, Hultgren SJ. Nat Rev Microbiol. 2020 (4):211-226. Flores-Mireles AL et al. Urinary tract infections: epidemiology, mechanisms of infection and treatment options. Nat Rev Microbiol. 2015 (5):269-84) have already addressed the topic, I believe that this work add valuable information and updates about this theme.

However, the authors did not properly discuss some topics of the review and a more detailed information about the pathogenesis of disease, the immune response during UTIs, and the treatment is missing.

Please, find below my corrections and suggestions:

·      Major corrections:

In the introduction, please discuss about the host susceptibility (i.e., the risk factors) associated with UTIs.

Line 48: please, describe or give some examples of which organisms (fungi and viruses) can also cause UTI.

Line 54: please, describe the complicating factors that are involved in the progression of disease severity.

In the introduction, add a paragraph discussing about the pathogenesis of disease and mention the figure 1 in the text.

Figure 1: please, improve the quality of figure, since it presents low resolution (especially the scheme of urinary tract).

Please, add a topic discussing about the immune response to UTIs.

Topic 3: Virulence factors of the main uropathogens: please, rewrite this topic, because it is repetitive and confusing.  

Lines 148 – 150: please, discuss about the function of these virulence factor in the pathogenesis of UTIs.

Line 202 - 203: describe the function of each component.

Lines 224- 226: please detail the treatment for UTIs, discussing which antibiotic is used depending on bacteria species (sensitive and resistant strains), disease, clinical manifestation. Also, authors could comment about drug combination. Authors can also add a table with a summary of UTIs treatment.

Line 231: the authors can elaborate more this sentence, bringing other references, such as:

Antimicrobial Resistance Collaborators. Global burden of bacterial antimicrobial resistance in 2019: a systematic analysis. Lancet. 2022 Feb 12;399(10325):629-655. doi: 10.1016/S0140-6736(21)02724-0. Epub 2022 Jan 19. Erratum in: Lancet. 2022 Oct 1;400(10358):1102. PMID: 35065702; PMCID: PMC8841637.

Figure 2: please, improve the quality of the figure since it presents a low resolution. Also, I suggest the author to add the management (e.g. therapeutic approach) applied for each type of UTI. Please, mention figure 2 in the text.

Topic 4: Antimicrobial resistance in UTI: I strongly suggest the authors to discuss about other mechanisms related to antimicrobial resistance, such as the alteration of bacterial proteins/enzymes that are antimicrobial targets and changes in membrane permeability to antibiotics (including the overexpression of ABC transporters).

·      Minor changes:

Please, check and correct the number of the topics.

Line 17: please, replace MDR by multidrug resistance (MDR)

Line 19: please, replace too few to very few

Line 32 – 34: please, replace (e.g., in the United States alone, it has been estimated that the economic burden of recurrent UTIs is more than $5 billion each year) to (e.g., it has been estimated that the economic burden of recurrent UTIs in the United States is more than $5 billion each year)

Table 1:

·      please add spaces between table lines;

·      In type 2, P fimbriae (binds to Globosides, a sub-class of the lipid class glycosphingolipid and promote the adhesion to plastic and glass), remove the period at the end of the sentence.

Topics 2.1 – 2. 5:

·      renumber for 3.1 – 3.5;

·      remove “adherence and invasion” from the titles.

Line 128: replace cytoplasm to cell cytoplasm.

Line 187: replace Catheter-associated urinary tract infections by Catheter-associated urinary tract infections (CAUTI)

Line 343: I suggest the authors to move the risk factors associated with UTIs to the introduction.

Author Response

Reviewer #1

We would like to thank the the reviewer since his/her comments have greatly helped us to improve the manuscript which was carefully revised according to his/her suggestions. Changes are shown in yellow.

  1. In the introduction, please discuss about the host susceptibility (i.e., the risk factors) associated with UTIs.

Thank you for the suggestion. This point is discussed in more detail in the revised manuscript (lines 73-76).

  1. Line 48: please, describe or give some examples of which organisms (fungi and viruses) can also cause UTI.

Thank you for the suggestion. Please see lines 49-51 of the revised manuscript.

  1. Line 54: please, describe the complicating factors that are involved in the progression of disease severity.

Thank you for the suggestion. Please see lines 73-79 of the revised manuscript.

  1. In the introduction, add a paragraph discussing about the pathogenesis of disease and mention the figure 1 in the text.

Thank you for the suggestion. A new paragraph was added in the revised manuscript (lines 58-689.

  1. Figure 1: please, improve the quality of figure, since it presents low resolution (especially the scheme of urinary tract).

Thank you for noting this. A higher resolution image has been uploaded

  1. Please, add a topic discussing about the immune response to UTIs.

Thank you for the suggestion. A new paragraph was added in the revised manuscript (lines 114-123).

  1. Topic 3: Virulence factors of the main uropathogens: please, rewrite this topic, because it is repetitive and confusing.  

The paragraph has been rewritten as requested. Hopefully, it is now clearer and more readable. Thanks again to the reviewer for the suggestion

  1. Lines 148 – 150: please, discuss about the function of these virulence factor in the pathogenesis of UTIs.

Thank you for the suggestion. The function of these virulence factors is now described in the revised manuscript (lines 200-207).

  1. Line 202 - 203: describe the function of each component.

Thank you for the suggestion. The function of these factors is now described in the revised manuscript (lines 274-280).

  1. Lines 224- 226: please detail the treatment for UTIs, discussing which antibiotic is used depending on bacteria species (sensitive and resistant strains), disease, clinical manifestation. Also, authors could comment about drug combination. Authors can also add a table with a summary of UTIs treatment.

The information has been added in the revised manuscript (lines 297-339).

  1. Line 231: the authors can elaborate more this sentence, bringing other references, such as: Antimicrobial Resistance Collaborators. Global burden of bacterial antimicrobial resistance in 2019: a systematic analysis. Lancet. 2022 Feb 12;399(10325):629-655. doi: 10.1016/S0140-6736(21)02724-0. Epub 2022 Jan 19. Erratum in: Lancet. 2022 Oct 1;400(10358):1102. PMID: 35065702; PMCID: PMC8841637.

Thank you for the suggestion. We have added two new references and more details to the sentence (lines 342-344)

  1. Figure 2: please, improve the quality of the figure since it presents a low resolution. Also, I suggest the author to add the management (e.g. therapeutic approach) applied for each type of UTI. Please, mention figure 2 in the text.

Thank you for the suggestion. A higher resolution image has been uploaded. Evaluation and management of urinary tract infections have been added for each type of UTI (please see figure 3 of the revised manuscript).

  1. Topic 4: Antimicrobial resistance in UTI: I strongly suggest the authors to discuss about other mechanisms related to antimicrobial resistance, such as the alteration of bacterial proteins/enzymes that are antimicrobial targets and changes in membrane permeability to antibiotics (including the overexpression of ABC transporters).

The information has been added in the revised manuscript (lines 380-389).

  • Minor changes:

Please, check and correct the number of the topics.

Thank you for noting this. The number of topics has been corrected

Line 17: please, replace MDR by multidrug resistance (MDR)

Done (line 17)

Line 19: please, replace too few to very few

Done (line 29)

Line 32 – 34: please, replace (e.g., in the United States alone, it has been estimated that the economic burden of recurrent UTIs is more than $5 billion each year) to (e.g., it has been estimated that the economic burden of recurrent UTIs in the United States is more than $5 billion each year)

 Thank you for the suggestion (lines 33-34).

Table 1:

  • please add spaces between table lines;
  • In type 2, P fimbriae (binds to Globosides, a sub-class of the lipid class glycosphingolipid and promote the adhesion to plastic and glass), remove the period at the end of the sentence.

The changes indicated by the reviewer have been included in Table 1. 

Topics 2.1 – 2. 5:

  • renumber for 3.1 – 3.5;
  • remove “adherence and invasion” from the titles.

Done. Thank you for the suggestion

Line 128: replace cytoplasm to cell cytoplasm

Done (line 178)

Line 187: replace Catheter-associated urinary tract infections by Catheter-associated urinary tract infections (CAUTI)

Done (line 227)

Line 343: I suggest the authors to move the risk factors associated with UTIs to the introduction.

Thank you for the suggestion.

Reviewer 2 Report

Urinary tract infections are one of the most common infectious diseases caused by bacteria, fungi, and viruses. This review examines recent advances in the pathogenesis of uropathogenic bacteria and discusses current and developing treatments. The manuscript is well-written and organized.

1.     There are minor editorial mistakes, such as the use of abbreviations (MDR, rUTI, etc) without prior definition, which should be corrected by providing a list of abbreviations in alphabetical order. Additionally, the term "UTIs" should be changed to "cUTIs" in line 38 to be consistent with the rest of the manuscript. Other editorial mistakes should also be checked.

2.     The authors should briefly discuss the similarities and differences in the adhesion and invasion of the main uropathogens to provide a more comprehensive understanding of the topic.

3.     The CFU/ml value in Figure 2 is not consistent with the description in the manuscript, which should be corrected to ensure the accuracy of the data.

4.     Line 279-285 should be discussed in section 3, which is related to the diagnosis and treatment of UTI.

5.     Bacteriophage therapy as a potential treatment for UTI is mentioned in the Discussion and Conclusion section, but it should also be discussed in section 3 related to the diagnosis and treatment of UTI.

6.     The manuscript "Li L, Li Y, Yang J, Xie X, and Chen H (2022) The immune responses to different Uropathogens call individual interventions for bladder infection. Front. Immunol. 13:953354. doi: 10.3389/fimmu.2022.953354" should be cited to provide more recent and relevant research on the topic.

Author Response

We would like to thank the the reviewer since his/her comments have greatly helped us to improve the manuscript which was carefully revised according to his/her suggestions. Changes are shown in yellow.

  1. There are minor editorial mistakes, such as the use of abbreviations (MDR, rUTI, etc) without prior definition, which should be corrected by providing a list of abbreviations in alphabetical order. Additionally, the term "UTIs" should be changed to "cUTIs" in line 38 to be consistent with the rest of the manuscript. Other editorial mistakes should also be checked.

Thank you for the suggestion. The indicated error has been corrected and a list of abbreviations has been included in the revised manuscript.

  1. The authors should briefly discuss the similarities and differences in the adhesion and invasion of the main uropathogens to provide a more comprehensive understanding of the topic.

We thank the reviewer for raising these points. These issues are now discussed more clearly in the revised manuscript (lines 131-143).

  1. The CFU/ml value in Figure 2 is not consistent with the description in the manuscript, which should be corrected to ensure the accuracy of the data.

Thank you for noting this. The error has now been corrected.

  1. Line 279-285 should be discussed in section 3, which is related to the diagnosis and treatment of UTI.

This point is now discussed in Section 3, as suggested by the reviewer (lines 338-343).

  1. Bacteriophage therapy as a potential treatment for UTI is mentioned in the Discussion and Conclusion section, but it should also be discussed in section 3 related to the diagnosis and treatment of UTI.

We agree with the reviewer and this point is now discussed in more detail in section 3 of the revised manuscript (lines 351-354).

  1. The manuscript "Li L, Li Y, Yang J, Xie X, and Chen H (2022) The immune responses to different Uropathogens call individual interventions for bladder infection. Front. Immunol. 13:953354. doi: 10.3389/fimmu.2022.953354" should be cited to provide more recent and relevant research on the topic.

Thank you for this important suggestion. An topic has been added to the revised manuscript (line 114-123).

Reviewer 3 Report

Major comments

There are many review articles in the field of UTIs and their treatment challenges and the role and mechanism of different bacteria in these infections. Authors should explain the importance of this article and its innovation and difference from other articles in the introduction section.

Tabel-1 needs to an appropriate title. In addition, some abbreviation used in the table should be defined. Other virulence factors involving in UTIs pathogenesis such as toxins, capsules, biofilms, QS and Iron Iron scavenging factors by different bacteria should be added to the Table-1.

Information related to the virulence factors of some important bacteria such as staphylococci species is not included in the table and text.

The role and frequency of different pathogens in developing UTIs should be presented in charts for both uncomplicated and complicated UTIs.  

The role of Iron scavenging factors and urease plays important role in the developing UTIs by different microorganisms. The role of these factor in the pathogenicity of UTIs and their presence in different microorganism have not been discussed.   

The use and accuracy of new detection methods such as molecular methods and MALDI-TOF MS should be discussed in the article.

In section of antibiotic resistance in UTIs the authors have been summarized the antimicrobial resistance among Gram negative bacteria particularly Enterobacteriaceae. They should included the antimicrobial resistant and therapeutic option of Gram positive bacteria such as VRE and MRSA.

In this section, the authors have only explained the mechanisms of resistance to beta-lactams, while resistance to other antibiotics has not been discussed. Also, the role and frequency of important factors in developing MDR such as integrons and efflux pumps have not been discussed.

In section of antibiotic resistance in UTIs, the authors need to discuss the level of resistance to each of the antimicrobial agents in different geographical areas and summarize this information in a table.

The importance of biofilms in UTIs and therapeutic approaches and strategies for biofilms associated UTIs should be discussed.

Combination therapy for UTIs caused MDR bacteria (both Gram negative and Gram negative) should be discussed with more detail and the results of combination therapy from different studies should be presented in Table.

The advance in the vaccine development for UTIs (Vaccines targeting bacterial adhesion, toxin and siderophore) should be disused discussed in more detail.

Minor comments

Abstract

Line 11. Phenotype is not appropriate for describing clinical signs, please change it to “symptoms”    

Line11-12; Urinary tract infection should be written in the abbreviated form (UTIs)

The name of bacteria should be written in the full and italic forms.

Text

Line-43 and 44 - The names of bacteria (Enterobacteriaceae and Escherichia coli) should be written in the full and italic forms.

Figure-1 in the article should be replaced with a more appropriate and high-quality figure. The figure should show the position of the urinary tracts in relation to other nearby tissues and organs.

Line-96 Uropathogenic Escherichia coli should be written in the abbreviated forms.

Line 98-100 (and other section of Manuscript) The name of bacteria should be written in italic forms.

Line 216 CAUTIES should be used in its full form.

Author Response

We would like to thank the the reviewer since his/her comments have greatly helped us to improve the manuscript which was carefully revised according to his/her suggestions. Changes are shown in yellow.

  1. There are many review articles in the field of UTIs and their treatment challenges and the role and mechanism of different bacteria in these infections. Authors should explain the importance of this article and its innovation and difference from other articles in the introduction section.

The contribution of this review article to the already existing literature is now included in the revised manuscript (lines 52-55)

  1. Tabel-1 needs to an appropriate title. In addition, some abbreviation used in the table should be defined. Other virulence factors involving in UTIs pathogenesis such as toxins, capsules, biofilms, QS and Iron Iron scavenging factors by different bacteria should be added to the Table-1.

A title has been added to Table 1 and a list of abbreviations has been defined. We agree with the reviewer that there are other virulence factors involved in the pathogenesis of UTI. These factors were not included in the table because our intent was to draw the reader's attention specifically to the adhesion factors expressed by uropathogens. If the reviewer agrees, we would like to avoid modifying the original table by including virulence factors other than adhesins. Please note that the virulence factors indicated are described in other sections of the revised manuscript (lines 151-153, 204-211, 437-443).

  1. Information related to the virulence factors of some important bacteria such as staphylococci species is not included in the table and text.

Thank you for noting this. This information has been added in the text and the table (table 1 and lines 257-268).

  1. The role and frequency of different pathogens in developing UTIs should be presented in charts for both uncomplicated and complicated UTIs.

We thank the reviewer for the comments, which we have addressed in the revised manuscript (please see figure 2 and lines 131-132 of the revised manuscript)

  1. The role of Iron scavenging factors and urease plays important role in the developing UTIs by different microorganisms. The role of these factor in the pathogenicity of UTIs and their presence in different microorganism have not been discussed.  

We thank the reviewer for the comments, which we have addressed in the revised manuscript (please see lines 230-238, 243-244, 264-268, 439-442)

  1. The use and accuracy of new detection methods such as molecular methods and MALDI-TOF MS should be discussed in the article.

We thank the reviewer for the comments, which we have addressed in the revised manuscript (please see lines 301-307)

  1. In section of antibiotic resistance in UTIs the authors have been summarized the antimicrobial resistance among Gram negative bacteria particularly Enterobacteriaceae. They should included the antimicrobial resistant and therapeutic option of Gram positive bacteria such as VRE and MRSA.

We thank the reviewer for the comments, which we have addressed in the revised manuscript (please see lines 407-413)

  1. In this section, the authors have only explained the mechanisms of resistance to beta-lactams, while resistance to other antibiotics has not been discussed. Also, the role and frequency of important factors in developing MDR such as integrons and efflux pumps have not been discussed.

We thank the reviewer for the comments, which we have addressed in the revised manuscript (please see lines 386-394).

  1. In section of antibiotic resistance in UTIs, the authors need to discuss the level of resistance to each of the antimicrobial agents in different geographical areas and summarize this information in a table.

We agree with the reviewer that the level of resistance to each antimicrobial agent in different geographic areas is an important issue, but this topic is very complex and its treatment would require a separate review. In addition, it is known that the limited implementation activities of the antimicrobial stewardship program and the different definitions of MDR contribute greatly to the differences in resistance observed in different geographic areas. We have added a sentence to comment on this important issue (lines 405-406).

  1. The importance of biofilms in UTIs and therapeutic approaches and strategies for biofilms associated UTIs should be discussed.

We thank the reviewer for the comments, which we have addressed in the revised manuscript (please see lines 139-150)

  1. Combination therapy for UTIs caused MDR bacteria (both Gram negative and Gram negative) should be discussed with more detail and the results of combination therapy from different studies should be presented in Table.

We thank the reviewer for the comments, which we have addressed in the revised manuscript (please see lines 309-345).

  1. The advance in the vaccine development for UTIs (Vaccines targeting bacterial adhesion, toxin and siderophore) should be discussed in more detail.

We thank the reviewer for the comments, which we have addressed in the revised manuscript (please see lines 437-453)

Minor comments

Line 11. Phenotype is not appropriate for describing clinical signs, please change it to “symptoms” 

Done

Line11-12; Urinary tract infection should be written in the abbreviated form (UTIs)

Done

The name of bacteria should be written in the full and italic forms

Done

Text

Line-43 and 44 - The names of bacteria (Enterobacteriaceae and Escherichia coli) should be written in the full and italic forms.

Done

Figure-1 in the article should be replaced with a more appropriate and high-quality figure. The figure should show the position of the urinary tracts in relation to other nearby tissues and organs.

Done

Line-96 Uropathogenic Escherichia coli should be written in the abbreviated forms.

Thanks, the mistake has been corrected

Line 98-100 (and other section of Manuscript) The name of bacteria should be written in italic forms.

The name of the bacteria were checked throughout the manuscript

Line 216 CAUTIES should be used in its full form.

Done

The manuscript has been proofread thoroughly and we sincerely hope it will meet with your approval

Round 2

Reviewer 3 Report

 The authors have inserted all the comment requested in the previous stage of the review and added them to the manuscript.